# Agroforestry Systems of Cocoa (*Theobroma cacao* L.) in the Ecuadorian Amazon

Leider Tinoco-Jaramillo [1], Yadira Vargas-Tierras [1], Nasratullah Habibi [2], Carlos Caicedo [1], Alexandra Chanaluisa [1], Fernando Paredes-Arcos [1], William Viera [3], Marcelo Almeida [4] and Wilson Vásquez-Castillo [5,*]

1   Central Amazon Research Site (EECA), National Institute of Agricultural Research (INIAP), Joya de Los Sachas 220350, Ecuador; leider.tinoco@iniap.gob.ec (L.T.-J.); yadira.vargas@iniap.gob.ec (Y.V.-T.); carlos.caicedo@iniap.gob.ec (C.C.); alexandra.chanaluisa@iniap.gob.ec (A.C.); fjparedes.a@hotmail.com (F.P.-A.)
2   Faculty of Agriculture, Balkh University, Balkh 1702, Afghanistan; nasratullah.habibi14@gmail.com
3   Tumbaco Experimental Farm, Santa Catalina Research Site, National Institute of Agricultural Research (INIAP), Tumbaco 170902, Ecuador; william.viera@iniap.gob.ec
4   Ciencias Aplicadas, Universidad de Las Américas (UDLA), Redondel del Ciclista, Vía a Nayón, Quito 170124, Ecuador; marcelo.almeida.marcano@udla.edu.ec
5   Ingeniería Agroindustrial, Universidad de Las Américas (UDLA), Redondel del Ciclista, Vía a Nayón, Quito 170124, Ecuador
*   Correspondence: wilson.vasquez@udla.edu.ec; Tel.: +593-984-659-247

**Abstract:** Agroforestry systems in the Ecuadorian Amazon play a vital role in environmental conservation and the promotion of sustainable agriculture. Therefore, it is crucial to demonstrate the benefits of the associated species within these production systems. This study aimed to assess the impact of agroforestry systems on cocoa yield, carbon sequestration, earthworm presence, and the nutritional contribution of companion species linked to cocoa (*Theobroma cacao* L.) cultivation under agroforestry systems. The research was conducted at INIAP's Central Experimental Station of the Amazon using a randomized complete block design with three replications. The agroforestry arrangements were: (1) monoculture; (2) forest (*Cedrelinga cateniformis* Ducke); (3) fruit forest (*Bactris gasipaes* Kunth); (4) service (*Erythrina poeppigiana* (Walp.) O.F.Cook); and (5) forest + service (*E. poeppigiana* + *C. cateniformis*). The results indicated that agroforestry systems showed better results than the monoculture in terms of yield (532.0 kg ha$^{-1}$ compared to 435.4 kg ha$^{-1}$) and total stored carbon (33.0–42.0 t ha$^{-1}$ compared to 39.6 t ha$^{-1}$). Additionally, agroforestry systems provided higher levels of Mg, B, and Ca, contributing to both crop yield and the presence of earthworms. These findings suggest a positive influence of companion species, improving soil nutrition through biomass incorporation and promoting environmental benefits (carbon sequestration). Therefore, agroforestry systems will support sustainable cocoa production in the Ecuadorian Amazon.

**Keywords:** biomass; carbon storage; earthworm abundance; nutrients

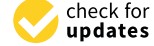



## 1. Introduction

Agriculture plays a key role in responding to growing global food demand [1]. However, current production patterns, together with social, economic, and demographic stresses, could lead to an unbalanced and unsustainable food supply [2]. To address this challenge, government interventions seek to promote technological advances that meet future needs in line with international trends [3] and that overcome biophysical limitations [4]. The predominant strategies focus on increasing land productivity through mechanization and greater dependence on external inputs [5], thus positioning public policies as drivers of agricultural development and the agrifood system [6].

In Ecuador, the agricultural sector represents approximately 10% of the gross domestic product and is a major source of employment (66%) of the economically active population in rural areas [7]. There have been many changes in the agricultural sector in the country that have caused economic instability for small, medium, and large producers. Agrarian reform initially focused on land distribution and land tenure, later evolving towards the implementation of programs supported by the state oriented towards research, technological innovation, and technology transfer, with financial support from international organizations [8]. The fragmentation of land, lack of continuity in projects, and focus on technological packages indicate the need to reconsider public policies to ensure a sustainable agrifood system [7].

In the Ecuadorian Amazon, there are political proposals that fail to ensure sustainable development but rather support oil and mining exploitation [9]. An alternative in the region is to promote sustainable production systems that promote food security and sovereignty [10] because the most appropriate land use in the Amazon is for forest and not for agricultural activities, let alone those related to conventional agriculture [11], due to adverse effects on natural resources. These include the reduction in biodiversity, soil erosion, and the alteration of ecosystems [12,13]. For this reason, sustainable production alternatives are currently being sought to reduce greenhouse gas emissions and help to improve crop productivity [14]. One option is the implementation of agroforestry systems, which contribute to diversification and environmentally friendly production in the long term, guaranteeing food security [15]. These systems associate multipurpose forest species with crops of agricultural interest [16]. They improve soil health by reducing the infiltration rate by 75% and runoff by 35%, decreasing erosion by 50%, increasing soil organic carbon by 21%, enhancing the storage of organic N by 13%, available nitrogen (N) by 46%, and 11% of phosphorus (P), and increasing soil pH by 2% [9]. The contribution of the biomass (pruning) of associated species improves soil quality, incorporating 150 to 300 kg of N, 10 to 20 kg of P, 75 to 150 kg of potassium (K), and 100 to 300 kg of calcium (Ca) per hectare and per year [17]. In addition, these systems conserve edaphic macrofauna [18], reduce the use of external inputs, improve profitability, support food security, and help the conservation and use of biodiversity [11]. Finally, they provide ecosystem services [15], such as habitat provision for wildlife, the conservation of animal and plant biodiversity, C storage, the regulation of hydrological cycles, and the recovery of degraded polluted areas, as well as climate change mitigation [11].

Promoting these sustainable production systems with cocoa (*Theobroma cacao* L.) is important because Ecuador produces 6% of the world's cocoa; 75% of its exports correspond to fine cocoa or national flavor, representing 60% of world production [19]. In 2022, exports of *T. cacao* and its processed products generated an income of USD 1005.7 million, representing 4.8% of total non-oil exports [20]. It is expected that the international price will remain above USD 2400 per ton during the next 4 years [21]. There are 591,556 hectares of *T. cacao* in the country, and 9.4% of this area is located in the Ecuadorian Amazon region (55,894 hectares), with 47.3% in Sucumbíos, 25.2% in Orellana, 24.1% in Napo, 0.2% in Pastaza, 1.2% in Morona Santiago, and 2.0% in Zamora Chinchipe [22]. In this region, high-yielding and pest-tolerant varieties are grown [23] in monoculture and agroforestry systems [22].

*T. cacao* is a crop that can be cultivated under agroforestry systems because it needs 50% shade for good growth and development [24], and so accompanying species must receive adequate silvicultural management [24,25]. In this region of Ecuador, agroforestry systems of *T. cacao* are characterized by their use for timber (*Cedrelinga cateniformis* Ducke, *Cordia alliodora* Ruiz & Pav., *Cedrela* spp., etc.), fruit (*Psidium guajava* L., *Inga* spp., *Bactris gasipaes* Kunth, etc.) [23], and services (*Erythrina* sp., etc.) [26]. The accompanying trees, especially leguminous plants, provide biomass, nutrients, and stored C in their aerial biomass; for example, it has been reported that the amount of C stored in a Chakra-type cacao agroforestry system (timber and non-timber forest species) is 141.4 Mg ha$^{-1}$ [27], and with *C. cateniformis* it is possible to store 2.82 to 14.29 Mg ha$^{-1}$ of C [28].

Del Águila Martínez [29] points out that the amount of C stored depends on the age of the accompanying trees and increases due to the passage of time, trunk diameter, height, and planting density [28]. In addition, [30] reported that in Central American countries, agroforestry systems of *T. cacao* with mixed shade (the forest, palm, and fruit species *Persea americana* Mill., *Pouteria sapota* J., *Mangifera indica* L., *Inga* spp., and *Citrus* spp.) store 117 Mg ha$^{-1}$ of C. In Ghana, agroforestry systems with this crop store 61.73 and 52.57 Mg ha$^{-1}$ of C in the soil and foliage, respectively, while monocultures store 5.98 and 5.73 Mg ha$^{-1}$ [31]. However, negative relationships can also arise; for example, companion species compete for light, nutrients, and water [32].

Besides storing and sequestering carbon, agroforestry systems can also potentially influence belowground soil biodiversity [33]. This diversity includes organisms such as earthworms, which are one of the most important soil biota because they influence the physical, chemical, and biological properties of soil [34,35]. Their movements create pores that facilitate the dynamics of nutrients and water in soil [36]. Earthworm populations and diversity vary between terrestrial habitats due to variations in soil moisture, soil temperature, soil properties, aboveground biomass abundance, vegetation types, land use management, and human interventions [37,38]. For example, in an agroforestry system with *Solanum quitoense* L. and *Selenicereus megalanthus* (Haw.) Britton & Rose in the Ecuadorian Amazon, it was found that earthworm abundance varied from 73 to 114 and 70 to 100 individuals, respectively, and, in a monoculture, from 42 to 67 and 51 to 91 individuals, respectively, and, that earthworm abundance depends on the sampling season, with more earthworms observed in the rainy season than the dry season [18,39].

In the Ecuadorian Amazon, monoculture cultivation continues to expand significantly, but currently, there is a trend to promote sustainable agriculture to conserve the region's resources. Little research has been performed concerning the agroforestry approach in crops of economic interest in this region [11]. Those long-term studies that do exist help to answer questions, such as if the yield of the main crop is affected by the presence of the associated species and if they play a key role in the contribution of nutrients, support macrofauna, and improve carbon sequestration. Consequently, the main objective of this study was to evaluate the influence of agroforestry systems on cocoa crop as well as its environmental benefits in comparison to a monoculture in a long-term analysis. The specific objectives were to evaluate crop yield, carbon storage, earthworm presence (soil macrofauna), and the nutritional contribution of companion species.

## 2. Materials and Methods

### 2.1. Experimental Site

The study was conducted at the Central Experimental Station of the Amazon (EECA) within the National Institute of Agricultural Research (INIAP). This is situated in the canton of La Joya de los Sachas, Orellana province, in the northern sector of the Ecuadorian Amazon region. The experimental site is located at 00°21′29.32′′ S and 76°51′47.76′′ W (Figure 1), at 250 masl. The climate is classified as humid subtropical, with abundant rainfall throughout the year: from 2700 to 3870 mm yr$^{-1}$. The mean annual temperature is 25 $\pm$ 3 °C, which corresponds to the Köppen–Geiger climate classification of "Af" (humid tropical climate) [40]. The ecosystem type is the lowland evergreen forest of Aguarico–Putumayo–Caquetá [41]. The soil type is classified as Andean Dystrudepts, with 43% clay, 27% silt, and 30% sand [42]. The experimental site was slightly acidic (pH: 5.6) and had good drainage, an organic matter content of 4 to 6%, and a slope of <2% [43].

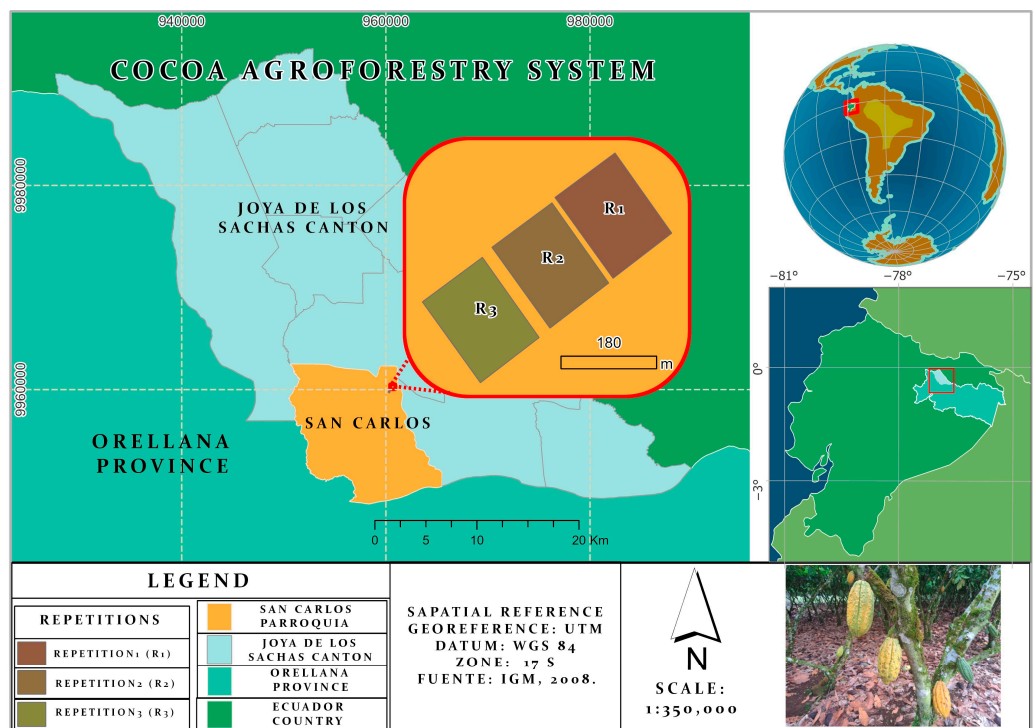

**Figure 1.** The cacao agroforestry system trial is situated in the canton of La Joya de los Sachas, Orellana, Orellana province.

### 2.2. Experimental Treatments

The experiment was structured with a completely randomized block design with five treatments and three replications, with a total of 15 experimental units. The experimental units consisted of 144 *T. cacao* plants. Thirty-six central plants were considered as observational units for the corresponding measurements and analysis. The total plot size was 1296 m². The agroforestry arrangements (agroforestry systems and monoculture) corresponded to: (1) monoculture; (2) forest with *C. cateniformis*; (3) fruit forest with *B. gasipaes*; (4) service with *E. poeppigiana*; and (5) forest + service with *E. poeppigiana* + *C. cateniformis*. All arrangements interacted with the experimental units of *T. cacao*. The evaluation was conducted over 5 years (2018 to 2022).

### 2.3. Crop Management

The study was implemented in a 3-year-old *T. cacao* agroforestry system using the species listed in Table 1.

**Table 1.** Species used in agroforestry systems, planting distance, and crown shape.

| Species | Use | Sowing Distance | Cup Shape |
|---|---|---|---|
| *T. cacao* | Fruit tree | 3 m × 3 m | Ellipsoidal [44] |
| *C. cateniformis* | Forestry | 12 m × 12 m | Rounded [45] |
| *B. gasipaes* | Fruit tree | 12 m × 12 m | Palemiform [46] |
| *E. poeppigiana* | Service | 6 m × 6 m | Oval [47] |

In 2018, 50% of the *E. poeppigiana* plants were pruned at a height of 2 m, while the other half were pruned at 4 m to also obtain a shade effect [48]. From 2019 to 2022, pruning consisted of removing 50% of the biomass. In the systems with *C. cateniformis*, *B. gasipaes*, and *E. poeppigiana*, biomass was incorporated every 180 days, while in the *E. poeppigiana* system, this was performed every 120 days [15]. The biomass from the pruning was cut into small pieces and spread on the soil surface at the crown of the *T. cacao* plants,

following the recommendations of Vargas-Tierras et al. [15]. For the *T. cacao* crop, pruning consisted of removing unproductive stems to improve air circulation and reduce the spread of pathogens in the crop.

For fertilization, ammonium nitrate (34% N), potassium nitrate (13% N and 46% K), monopotassium phosphate (52% P and 34% K), magnesium nitrate (10% N and 15% Mg), active yaramila (20% N, 7% P and 10% K), and yaramila complex (12.4% N, 11% P, 18% K and 2.7% Mg) were used. In total, 76 to 134 g plant$^{-1}$ yr$^{-1}$ of N, 94 to 134 g plant$^{-1}$ yr$^{-1}$ of P, and 120 to 134 g plant$^{-1}$ yr$^{-1}$ of K were used. Phytosanitary controls for *Phytophthora palmivora* and *Moniliophtora roreri* were carried out every 21 days during the reproductive phase (180 days) of the crop. It is important to note that almonds are protected by the shell of the cob until harvest, and thus contamination by pesticides is unlikely [49]. Weed control was performed using rotating glyphosate (4.5 cc lt$^{-1}$) and 2–4 D amine (2.3 cc lt$^{-1}$) every 120 days. In terms of irrigation, the northern part of the Ecuadorian Amazon has a rainfall of 3050 to 3100 mm per year, which supplies the water requirements of the cocoa crop.

*2.4. Study Variables*

2.4.1. Cocoa Yield

Cocoa yield was obtained by weighing all of the harvested cocoa beans and was expressed in kg plant$^{-1}$. The yield obtained per treatment was extrapolated to fresh kilograms per hectare and multiplied by a constant (0.40) to obtain the dry cocoa yield in kg ha$^{-1}$ yr$^{-1}$ [50].

2.4.2. Concentration of Nutrients in Leaf Biomass

To determine the N, P, K, Ca, Mg, S, Fe, B, Cu, and Zn content, the total biomass generated in each treatment was multiplied by the dry matter produced by each legume species; subsequently, the equation proposed by [51] for macro and micro elements was applied.

$$Q = \frac{[MST * X]}{10^2}$$

$Q$ = Amount of nutrient present in total dry matter (kg nutrient ha$^{-1}$)
$MST$ = Total dry matter
$X$ = Concentration of nutrients in dry matter

The nutrient content of each treatment was extrapolated to estimate the total amount of N, P, K, Ca, Mg, S, B, Zn, Fe, and Cu in kg per hectare per year. The Kjeldahl method was used to determine the total N [52,53]. For P determination, the nitric–perchloric digestion extract colorimetric method was used, while K, Ca, Mg, S, S, B, Zn, Fe, and Cu were determined via atomic absorption spectrometry [52].

2.4.3. Soil Nutrient Concentration

From 2018 onwards, composite soil samples (1 kg) were collected in labeled plastic bags. Soil sampling was performed at a depth of 20 cm. Soil organic carbon content was determined, and total N, P, K, Ca, Mg, and S were expressed as percentages and Fe, B, Cu, and Zn as ppm. N quantification was carried out via the Kjeldahl method [52,53], P was determined by colorimetry, K, Ca and Mg were evaluated by atomic absorption spectrometry, and S was quantified by turbidimetry. Carbon (C) was determined via the Walkey and Black method [52].

2.4.4. Estimation of Carbon Storage

Starting in 2018, samples of three trees from each type of multipurpose species were collected from each treatment to quantify the biomass obtained by pruning. The total amount of biomass from pruned leaves and branches was measured at the collection site using a digital scale (Camry, model ASC-30-JC11B, Jiangsu Sheng, China) and the weight in kilograms was re-recorded. The mean biomass of the examined trees was utilized to calculate the overall fresh biomass per hectare by multiplying this figure by the total number

of trees [54]. To establish the total C storage in the systems, the values of carbon stored in the aerial biomass, including stem, coarse roots, and soil, were added together [30,55,56]. The C present in the biomass was considered to be 50% when following the method proposed by [57]. Allometric variables were used to determine the C in the biomass of the stem and coarse roots (Table 2). The density value of the wood from *C. cateniformis* (450 kg m$^{-3}$) and *E. poeppigiana* (205 kg m$^{-3}$) was taken into account [58]. The stem diameter at breast height was measured at 1.30 m from the ground, and total stem height was evaluated every year. From these data, the biomass per individual and per year were calculated. These results were then multiplied by the number of plants per hectare to obtain the value in kg ha$^{-1}$.

**Table 2.** Allometric variables for the estimation of biomass in the stem and root of the forest species evaluated during the study.

| Species | Equation |
| :---: | :---: |
| *B. gasipaes* | $B = 6.66 + 12.826 * H^{0.5} * Ln(H)$ [55] |
| *C. cateniformis* | $B = e^{(-2.231+0.933*Ln(DBH^2*H*\rho))}$ [57] |
| Root biomass | $BS = e^{(-1.0587+0.8836*Ln(B))}$ [58] |

where: $B$ = aboveground biomass (kg ha$^{-1}$); $DBH$ = diameter at breast height (m); $H$ = total plant height (m); $\rho$ = wood density (kg m$^{-3}$); $BS$ = belowground coarse root biomass (kg ha$^{-1}$).

For the determination of soil C storage, the organic C from the soil analysis was taken into account and the following formula was applied:

$$SOC = SOF \times Da \times P \times 100$$

where: $SOC$ = soil organic carbon (t ha$^{-1}$); $SOF$ = soil carbon fraction (%); $Da$ = bulk density (0.8 t m$^{-3}$); $P$ = sampling depth (0.20 m); and 100 = constant for transformation to t ha$^{-1}$ [43].

2.4.5. Number of Earthworms

The abundance of earthworms (*Eisenia* sp.) was assessed during their peak biological activity, which occurred in April (rainy season) and October (dry season), at each sampling plot. An area 0.5 m long × 0.5 m wide was delimited, and we sampled a total of 1.0 m$^2$ at a depth of 0.20 cm. The sampling process was conducted twice, with two samplings executed within the cocoa plant rows and two in the center of these rows [18].

*2.5. Data Analysis*

R software, version 4.1.2, was used in this study to perform statistical data analysis. Analysis of variance (ANOVA) was conducted to evaluate the impact of the different treatments and production cycles. Tukey's test was used to perform significant comparisons between treatment means.

Explanatory variables were selected using correlation analysis. This method facilitated the identification of optimal combinations of explanatory variables that closely correlated with crop yield. Furthermore, correlation calculations were conducted to scrutinize the connection between soil minerals and the quantity of earthworms present during each production cycle. This analytical approach aimed to unveil the interplay between these influential factors and enhance our understanding of their collective impact on agricultural productivity.

**3. Results**

*3.1. Cocoa Yield*

The analysis showed that cocoa production cycles ($p < 0.0001$) had a highly significant effect on treatments, but not on interaction ($p = 0.9999$) (Table 3). The main effect for yield across production cycles showed that the yield in 2018 was higher (697.2 kg ha$^{-1}$) than

in subsequent cycles. Yield decreased by 44% in 2019, then increased by 18% in 2020, then decreased by 9% in 2021, and finally increased by 16% in 2022 (Figure 2). The main effect on the agroforestry systems was that the dry cocoa yield was higher in the combined system (*C. cateniformis* + *E. poeppigiana*). In addition, it was determined that cocoa grown under agroforestry systems had higher yields than that grown in monoculture conditions (Table 4).

**Table 3.** Statistical significance of individual factors and interaction effect on *T. cacao* yield.

| Factors | Yield (kg ha$^{-1}$) |
| --- | --- |
| Production cycles | ** |
| Agroforestry systems | * |
| Production cycles × agroforestry systems | NS |

NS: not significant; *: significant at $p \leq 0.05$; **: significant at $p \leq 0.01$.

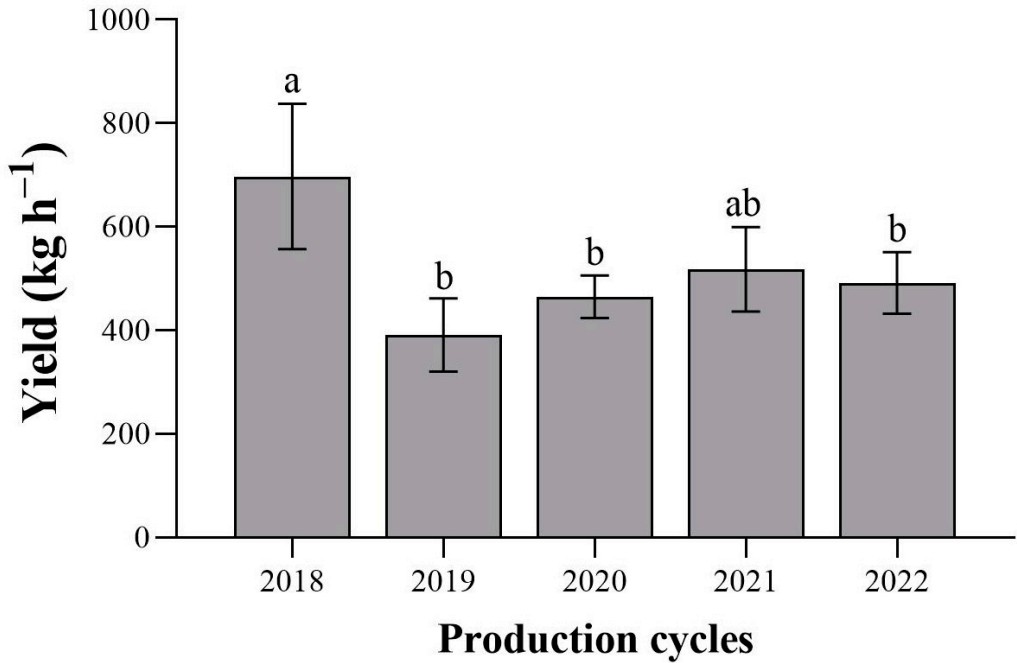

**Figure 2.** Mean values of dry cocoa yield in the different production cycles. Different letters indicate significant differences ($p < 0.05$) using a one-way ANOVA analysis followed by Tukey's Test.

**Table 4.** Mean values for fruit yield determined by the agroforestry systems. Mean values are reported. Within a column and within a given factor, means followed by the same letter are not statistically different ($p < 0.05$).

| Agroforestry System | Yield (kg ha$^{-1}$) |
| --- | --- |
| *E. poeppigiana* | 575.3 ab |
| *C. cateniformis* + *E. poeppigiana* | 603.9 a |
| *C. cateniformis* | 510.4 b |
| *B. gasipaes* | 438.3 b |
| Monoculture | 435.4 c |

In the production cycles of 2018, 2019, 2021, and 2022, the dry cocoa yield demonstrated an increase in both the combined system (*C. cateniformis* + *E. poeppigiana*) and the system with *E. poeppigiana* when compared to the monoculture. However, in 2020, the monoculture cocoa system exhibited a 7% higher yield than the system with *B. gasipaes* (Table S1).

### 3.2. Nutrients Influencing Cocoa Yield

The analysis of soil nutrients revealed intricate correlations between different elements. A slightly negative correlation of −0.57 existed between Mn and Ca, indicating an inverse relationship. Cu also showed negative correlations of −0.51, −0.54, and −0.52 with Mo, N, and C, respectively. Additionally, there was a significant positive correlation of 0.69 between Zn and Cu, indicating a simultaneous increase in their levels. N also displayed strong positive correlations of 0.94 with both Mo and C. These results (Figure 3) provide valuable insights into the complex interactions of soil nutrients, offering information for agricultural practices to enhance soil health and fertility optimization.

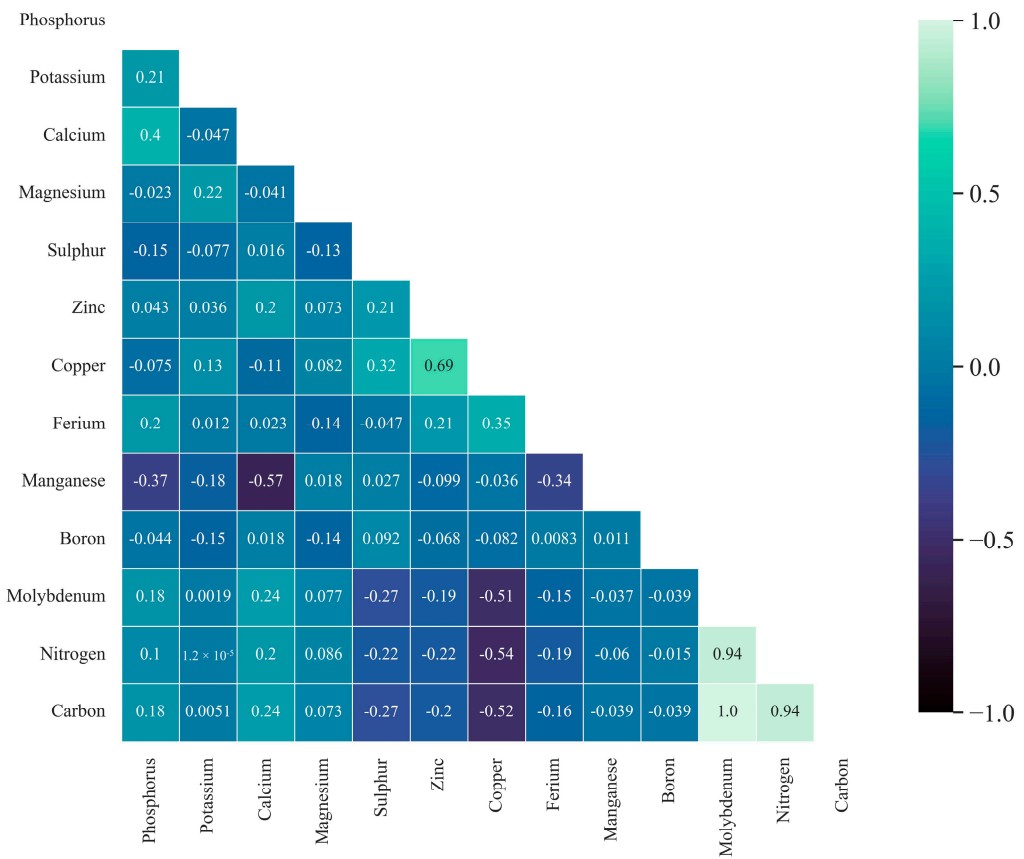

**Figure 3.** Correlation matrix of soil nutrients present in the experimental site.

The nutrients with the greatest impact on cocoa yield were Mg and B, although these elements did not exhibit statistical significance. Furthermore, there was an inverse correlation between this parameter and Fe, K, Ca, and S (Table 5).

**Table 5.** Results of correlation analysis to identify nutrients that contributed to crop yield.

| Nutrient | Estimate | Probability |
|:---:|:---:|:---:|
| S | −3.6 | 0.5 NS |
| Mg | 33.7 | 0.7 NS |
| Fe | −0.9 | 0.1 * |
| B | 119.7 | 0.1 NS |
| K | −188.0 | 0.03 * |
| Ca | −20.0 | 0.04 * |

NS: not significant; *: significant at $p \leq 0.05$.

### 3.3. Total Carbon Stored

The analysis showed that there was a significant effect between agroforestry systems and production cycles ($p$ = 0.08 and 0.0005,) but that there was no interaction ($p$ = 0.9999) (Table 6).

**Table 6.** Statistical significance of individual factors and interaction effect on total C storage.

| Factor | Total Stored C |
|---|---|
| Production cycle | * |
| Agroforestry system | * |
| Production cycles × agroforestry systems | NS |

NS: not significant; *: significant at $p \leq 0.05$.

C storage increased over time (Table 7). The agroforestry systems that stored the most C in biomass and soil were *E. poeppigiana* and the combined system *C. cateniformis + E. poeppigiana*, and the systems that stored the least C were those that had only one species (Table 8). The interaction between the agroforestry systems and the years of evaluation was not significant (Table S2).

**Table 7.** Mean values for stored C determined by production cycles. Within a column and within a given factor, means followed by the same letter are not statistically different ($p$ < 0.05).

| Production Cycle | Stored C Biomass (Aboveground and Roots) (t ha$^{-1}$) | Stored C in the Soil (t ha$^{-1}$) | Total Stored C (t ha$^{-1}$) |
|---|---|---|---|
| 2022 | 2.2 b | 43.9 c | 47.2 a |
| 2021 | 2.1 b | 40.0 bc | 44.8 a |
| 2020 | 2.1 b | 32.0 a | 36.5 b |
| 2019 | 1.6 a | 28.4 a | 33.1 b |
| 2018 | 1.6 a | 34.0 ab | 36.3 b |

**Table 8.** Mean values for stored C determined by agroforestry system. Within a column and within a given factor, means followed by the same letter are not statistically different ($p$ < 0.05).

| Agroforestry System | Stored C Biomass (Aboveground and Roots) (t ha$^{-1}$) | Stored C in the Soil (t ha$^{-1}$) | Total Stored C (t ha$^{-1}$) |
|---|---|---|---|
| *E. poeppigiana* | 3.6 c | 38.3 a | 42.0 b |
| *C. cateniformis + E. poeppigiana* | 1.9 b | 37.1 a | 39.1 ab |
| *C. cateniformis* | 1.7 b | 32.4 a | 34.1 a |
| *B. gasipaes* | 1.2 a | 31.7 a | 32.9 a |
| Monoculture | 1.1 a | 38.4 a | 39.6 ab |

### 3.4. Abundance of Earthworms

The analysis revealed significant statistical variances in the number of earthworms per agroforestry system during distinct sampling seasons, including both the rainy and dry seasons (Table 9). No noteworthy differences were observed based on the production cycle or in the interaction between sampling season and agroforestry system. The number of earthworms exhibited an increase in the combined system (*C. cateniformis + E. poeppigiana*) during the rainy season and in the system with *B. gasipaes* during the dry season (Table 10).

**Table 9.** Statistical significance of individual factors and the effect of the interaction on earthworm abundance.

| Factor | Earthworm Abundance | Season |
|---|---|---|
| Production cycle | NS | |
| Agroforestry system | * | Rainy season |
| Production cycles × agroforestry systems | NS | |
| Production cycle | NS | |
| Agroforestry system | * | Dry season |
| Production cycles × agroforestry system | NS | |

NS: not significant; *: significant at $p \leq 0.05$.

**Table 10.** Mean values for the number of earthworms for the agroforestry systems in the two evaluated seasons. Within a column and within a given factor, means followed by the same letter are not statistically different ($p < 0.05$).

| Agroforestry System | Season | Earthworm Abundance |
|---|---|---|
| *E. poeppigiana* | | 14.07 b |
| *C. cateniformis + E. poeppigiana* | | 28.17 a |
| *C. cateniformis* | Rainy season | 22.26 ab |
| *B. gasipaes* | | 23.93 ab |
| Monoculture | | 11.73 b |
| *E. poeppigiana* | | 17.13 b |
| *C. cateniformis + E. poeppigiana* | | 24.87 a |
| *C. cateniformis* | Dry season | 15.13 b |
| *B. gasipaes* | | 27.13 a |
| Monoculture | | 16.21 b |

*3.5. Nutrients Influencing Earthworm Abundance*

Soil nutrients that positively impacted the number of earthworms during the rainy season included Cu and Ca, although these elements lacked statistical significance. Conversely, in the dry season, the most influential elements were Ca, Zn, and N, with the latter two not exhibiting statistical differences (Table 11). On the other hand, B had a negative influence on the number of earthworms in the two evaluation seasons.

**Table 11.** Results of the correlation analysis to identify the nutrients that contributed to earthworm abundance.

| Season | Nutrient | Estimate | Probability |
|---|---|---|---|
| | Cu | 0.65 | 0.4730 NS |
| | B | −7.84 | 0.353 NS |
| Rainy season | Ca | 0.46 | 0.588 NS |
| | Zn | −1.15 | 0.031 * |
| | N | −53.63 | 0.147 NS |
| | Cu | −0.58 | 0.521 NS |
| | B | −7.57 | 0.369 NS |
| Dry season | Ca | 2.02 | 0.020 * |
| | Zn | 0.42 | 0.424 NS |
| | N | 12.47 | 0.733 NS |

NS: not significant; *: significant at $p \leq 0.05$.

The correlation analysis was carried out by considering soil nutrients and the number of earthworms in the different years of evaluation (Table 12). In 2018's rainy season, the relationship between the number of earthworms with Ca and B showed moderate inverse relationships (−0.69 and −0.57, respectively), while in 2021, Ca showed a slight correlation (0.52). In the dry seasons of 2018 and 2020, relatively high correlations (0.73, 0.60, and 0.75)

were found between B and Zn with the number of earthworms, respectively. In 2022, a moderate correlation with Ca (0.69) and a moderate inverse correlation ($-0.60$) with Mg were determined to exist.

**Table 12.** Correlation coefficients of nutrients with earthworms at different production cycles.

| Nutrient | Season | Years | | | | |
|---|---|---|---|---|---|---|
| | | **2018** | **2019** | **2020** | **2021** | **2022** |
| Earthworms vs. Ca | | $-0.69$ | $-0.39$ | $-0.38$ | 0.52 | 0.18 |
| Earthworms vs. Mg | Rainy | $-0.05$ | $-0.31$ | $-0.42$ | 0.33 | 0.23 |
| Earthworms vs. Zn | | $-0.41$ | $-0.45$ | $-0.37$ | 0.19 | 0.28 |
| Earthworms vs. B | | $-0.57$ | 0.28 | $-0.24$ | 0.16 | 0.13 |
| Earthworms vs. Ca | | 0.20 | 0.07 | 0.24 | 0.31 | 0.69 |
| Earthworms vs. Mg | Dry | 0.16 | 0.32 | 0.29 | 0.33 | $-0.60$ |
| Earthworms vs. Zn | | 0.30 | $-0.07$ | 0.54 | 0.75 | $-0.24$ |
| Earthworms vs. B | | 0.73 | $-0.24$ | $-0.61$ | 0.60 | 0.34 |

## 4. Discussion

### 4.1. Cacao Yield

In 2018, the dry cocoa yield was higher (697.2 kg ha$^{-1}$) than in subsequent years; this behavior was due to the facts that the cocoa trees were initiating their productive phase [59] and the canopy of the shade trees that made up the system had not yet closed. That is, it had open spaces (little shade), meaning that the incidence of pests and diseases was nil [60]. On the other hand, in 2019 and 2021, the production decreased by 44 and 9% with respect to each previous year and increased by 18% and 16% in 2020 and 2022 compared to 2019 and 2021, respectively. This fluctuation in yield may be because cocoa is considered a biennial crop, that is, there are years when yields are better than others, a hypothesis that was corroborated by Quiroga and Asante et al. [61,62], who mentioned that this crop had a well-marked biannual oscillation during its life cycle.

The main effect of agroforestry systems was determined to be that the dry cocoa yield was higher in the combined system (*C. cateniformis* + *E. poeppigiana*) and with *E. poeppigiana* (603.9 and 575.3 kg$^{-1}$ ha$^{-1}$ yr$^{-1}$). This behavior was also reported by Kouassi et al. [63], who stated that cocoa plantations intercropped with certain forest species that provide shade to the crop improved its productivity. The yield results obtained in this study are higher than the data (105 and 353 kg$^{-1}$ ha$^{-1}$ yr$^{-1}$) reported by Mensah et al. and Schneider et al. [64,65] for mixed agroforestry systems (fruit, timber, spices, and medicinal plants) of cocoa in Alto Beni (Bolivia) and Ghana. Our results are also higher than values reported for cocoa associated with 5-year-old *Cojoba arborea* (L.) Britton & Rose *and Dalbergia glomerata* Hens (500 and 458 kg$^{-1}$ ha$^{-1}$ yr$^{-1}$, respectively) [66] and 17-year-old cocoa systems associated with *Dipterix panamensis* (Pittier) Record & Mell, *Guarea grandifolia* DC., and *Terminalia superba Engl. & Diels* (430, 341, and 212 kg$^{-1}$ ha$^{-1}$ yr$^{-1}$) [67]. However, they are lower than the yield (765 kg$^{-1}$ ha$^{-1}$ yr$^{-1}$) reported by Ramírez-Argueta et al. [66] for cocoa agroforestry systems with *Guarea grandifolia*, *D. glomerata*, *C.arborea*, *Plathymiscium dimorphandrum* Donn. Sm., *Tabebuia donnell-smithii* Rose, *Ilex tectonica* W.J.Hahn, *Calophyllum brasiliense* Cambess, *Hyeronima alchorneoides* L., *Lonchocarpus* sp., *Nectandra* sp., *Macrohasseltia macroterantha* Standley & L. O. Williams., and *Swietenia macrophylla* King in Honduras, and for systems with *C. alliodora* and *E. poeppigiana* (626 and 712 kg$^{-1}$ ha$^{-1}$ yr$^{-1}$, respectively) in Costa Rica [68].

In general, it was determined that cocoa grown under agroforestry systems has higher yields than cocoa grown in monoculture; this same behavior was observed by Kouassi et al. [63], who mentioned that cocoa production increases under shade compared to full-sun cultivation (monoculture) because cocoa pods weigh more when grown in agroforestry systems. This finding differs from the results obtained by Scheneider et al. [64], who found that a monoculture produces more (587 kg ha$^{-1}$ yr$^{-1}$) than an agroforestry system (105 kg ha$^{-1}$ yr$^{-1}$). In addition, the production obtained in the different treatments of

this study exceeded the average yield reported in agroforestry systems in the provinces of Orellana and Napo (310 and 270 kg ha$^{-1}$ yr$^{-1}$, respectively) [69], as well as the national and world average production rates (300 and 480 kg ha$^{-1}$ yr$^{-1}$, respectively) [50]. This study demonstrates that cocoa grown under agroforestry systems is a sustainable production alternative because the yield is better than in monoculture. Above all, the hypothesis is accepted that growing cocoa under agroforestry systems is a viable way for the Ecuadorian Amazon to tackle climate change, which poses many difficulties for the future of agriculture because temperatures will be higher and more variable and there will be changes in rainfall and humidity [70], thus threatening production across the world.

### 4.2. Nutrients Influencing Cocoa Yield

The nutrients that were determined to be essential for cocoa production were Mg and B. Although plants do not require these elements in large quantities, they are indispensable for maintaining optimal growth and development [66]. To produce one metric ton of cocoa pods, 1.5 and 6 kg$^{-1}$ ha$^{-1}$ yr$^{-1}$ of Mg [68] and 4.4 g tree$^{-1}$ of B [69] are needed; this application generates an increase in yield of up to 180% [71]. On the other hand, B deficiency causes a 40% reduction in yield, with the presence of deformed fruits, short internodes, brittle leaves, small seeds, and increased susceptibility to diseases and viruses [72,73]. K, S, Fe, and Ca were not determinants of cocoa yield. This may occur when the concentrations of these elements are stable [71]. In addition, shade-grown cocoa does not require large amounts of K and Ca as when grown in monoculture [74]. Although the nutritional requirements of S and Fe in cocoa cultivation remain unknown, it is known that the amount of these elements in the soil depends on soil type and soil mineral sources [75]. It is important to note that the amount of nutrients incorporated into the system depends on the amount of biomass produced. In this study, the biomass incorporated per plant of *C. cateniformis*, *B. gasipaes*, and the combination *C. cateniformis + E. poeppigiana* ranged, on average, from 12 to 21 kg, 17 to 33 kg, and 10 to 13 kg, respectively; and in the *E. poeppigiana* system, between 7 and 18 kg plant$^{-1}$ of biomass per plant were incorporated between the second and fifth years.

### 4.3. Total Carbon Stored

This study determined that the total C stored (soil, biomass area, and coarse roots) in the different production cycles increased as the main crop matured (36.3 to 47.2 t ha$^{-1}$). These values were higher than the lowest value determined for 23.5-year-old agroforestry systems in Central American countries (47 t ha$^{-1}$) [30] and for agroforestry systems of 5 and 20 years of age (35 t ha$^{-1}$) in Ghana [76]. From the above, it could be inferred that the amount of C stored is associated with the age of the crop, composition of the accompanying species, density, and number of trees per hectare [77,78].

Furthermore, it was determined that the agroforestry systems with legumes (*E. poeppigiana*, *C. cateniformis + E. poeppigiana*, and monoculture) stored 38.3, 37.1, and 38.4 t ha$^{-1}$ of C in the soil, respectively, and the systems with *B. gasipaes* and *C. cateniformis* stored more C (31.7 and 32.4 t ha$^{-1}$, respectively). Although what happened in these two systems could not be clearly explained, it appears that the amount of C stored in the soil in the 5 evaluated systems exceeded the C contents stored in the soil in agroforestry systems with conventional and organic production (27 to 36 t ha$^{-1}$, respectively) [77]. Another possible explanation is that which Ber-hongaray et al. [79] mentioned, that soil C is less likely to undergo changes.

It was also found that *E. poeppigiana* and *C. cateniformis + E. poeppigiana* stored more C in the aerial region than in single-species systems. Similar results were reported for 15- and 30-year-old mixed shade cocoa systems located in eastern Ghana (3 t ha$^{-1}$); in contrast, in 11-year-old cocoa plots grown as a monoculture, C storage was 4 t ha$^{-1}$ [30]. Cocoa grown with 8-year-old *Cordia* sp. stored 6.0 and 3.0 t ha$^{-1}$ of C; and in plantations older than 8 years, C storage was 8.7 and 10.1 t ha$^{-1}$ [80]. It was also found that in both agroforestry systems with fruit trees (*B. gasipaes*) and monoculture, aerial C storage was 1.2 t ha$^{-1}$, a

value that exceeds that reported in agroforestry systems of cacao with banana (*Mussa* sp.) and avocado (*Persea* sp.), which specifically are 0.8 and 0.7 t ha$^{-1}$ [81], but lower than the C storage reported for aboveground biomass in agroforestry systems in Central America (1.3 t ha$^{-1}$) [30]. In general, Montagnini and Nair [82] point out that agroforestry systems with perennial crops are important carbon sinks, while agroforestry systems with annual crops are more similar to conventional agriculture. Given this statement, it can be said that in the short, medium, and long term, agroforestry systems are the most viable option for storing and sequestering C in the Ecuadorian Amazon.

### 4.4. Abundance of Earthworms

The abundance of earthworms in the rainy and dry seasons in the agroforestry systems was higher (22 and 21 individuals, respectively) than in the monocultures (12 and 14 individuals, respectively). This behavior indicates that the using agroforestry systems is a strategy capable of minimizing the impacts of climate change. This is because there are studies that show that when the temperature increases by 3 degrees Celsius, the humidity will decrease by 50 or 52% [83,84]. It was also determined that the abundance of earthworms was higher in the rainy season than in the dry season. Vršič et al. [85] also found higher earthworm abundance in the first half of the year than during the summer; thus, the presence of these macroinvertebrates was null at various times [18,86]. The highest abundance of earthworms was found in the systems with *C. cateniformis + E. poeppigiana* and *B. gasipaes*. These findings coincide with those stated by Vargas et al. [18], who argued that agroforestry systems have a higher abundance of earthworms than monocultures due to the contribution of biomass and the formation of microclimates. Dekemati et al. and Vršič et al. [85,87] reconfirmed that earthworms can grow and proliferate faster when they receive plant residues in considerable quantities.

Moreover, the sampling of earthworm abundance was performed at a depth of 0.20 m, where it was determined that the number of individuals was less than 30; these results exceeded those reported by Vršič et al. [85], who found 20 individuals in soils with vegetation cover and mulch at depths of 0.15 to 0.30 m. Furthermore, the variation in the number of earthworms in the different evaluation cycles was possibly due to the amount of biomass present on the soil surface and to environmental conditions. Dekemati et al. and Mulia et al. [87,88] noted that earthworm species diversity varies between terrestrial habitats because of variations in soil moisture, temperature, and properties as well as the abundance of surface cover, vegetation types, land use management, and human intervention. Therefore, earthworms are sensitive to land use change.

### 4.5. Nutrients Influencing Earthworm Abundance

The soil nutrient that positively influenced the number of earthworms was Ca in the two sampling periods. This possibly occurred because the accompanying species are leguminous plants that provide important amounts of Ca and Mg for both the crop and the soil macrofauna [18]. In addition, before each fertilization, lime was applied to regulate soil pH. Muvahhid [89] pointed out that lime is beneficial for many types of earthworms because it provides Ca, an element that is required for earthworm growth and multiplication [90]. Meanwhile, B negatively influenced the number of earthworms in the two evaluation seasons, although it was beneficial for cocoa crop production. Santos et al. [91] found that B in high concentrations limits the presence of earthworms due to toxicity in their cells [92].

Finally, Cu, Zn, and N behaved differently in the two evaluation periods; this variation may have occurred because Zn and Cu can be physiologically regulated by some earthworm species [93]. The negative effects of N, Cu, and Zn on the number of earthworms may occur because, according to the soil analysis reports, the contents are above those established as optimal (20 to 40, 1 to 4, and 2 to 7 ppm, respectively). In this study, the Cu and Zn contents were above 6.5 and 25 ppm, which could be considered high if compared to the parameters given for contaminated soils (6.5 to 15.3 and 25.2 to 96.8 ppm,

respectively) [94]. Other studies state that the use of agrochemicals causes the soil pH to change [94] to have unsuitable values (very acidic or very alkaline), causing adverse effects on soil biodiversity and biological activity [95]. Herbicide application may also influence earthworm abundance. Vršič et al. [85] noted that earthworm numbers are relatively low in crops where herbicide, especially glyphosate, is continuously applied.

Cocoa is one of the most significant agricultural crops worldwide, especially in Ecuador and even more so in the Ecuadorian Amazon where it is grown on 82% [9] of farms. It is considered a family farming crop that generates a significant source of income. Current agricultural yields in this region are low (310 kg ha$^{-1}$ yr$^{-1}$) and are expected to decrease in response to climate change. To improve yields and foster adaptation to climate change, it is crucial to understand the main drivers affecting on-farm productivity. For this reason, this study demonstrated that agroforestry systems have more benefits than monocultures in terms of yield, earthworm abundance, C storage, and nutrient supply.

The benefit–cost ratio (based on the implementation, production costs and yield) for the agroforestry system with *C. cateniformis* + *E. poeppigiana* was 1.47, while it was 1.05 for the monoculture. Therefore, an agroforestry system would be more profitable than a monoculture. However, a deeper long-term economic analysis must be carried out to have a full panorama of the performance over time of this sustainable technology in monetary terms.

In addition, it is important to note that the adoption of sustainable technologies is subject to a change in ideology on the part of the farmers and the implementation of the Ministry of Agriculture's policies and guidelines on sustainable production.

## 5. Conclusions

Cocoa grown under agroforestry systems had higher yields (532.0 kg ha$^{-1}$) than the monoculture (435.4 kg ha$^{-1}$), making it a sustainable option for farmers in the Amazon Region. The cocoa yields in the agroforestry systems were 42 and 27% higher than the average yield reported for the province of Orellana (310 kg ha$^{-1}$ yr$^{-1}$) and the national level (390 kg ha$^{-1}$ yr$^{-1}$). In addition, the results suggested that the shade provided by the accompanying species can influence cocoa productivity in the long term, thus delivering maximum benefits for climate change adaptation and mitigation in the Amazon region.

The cocoa agroforestry systems stored more total C (33.0 to 42.0 t ha$^{-1}$) than the monoculture (39.6 t ha$^{-1}$), thereby generating positive environmental impacts. Additionally, the contribution of biomass positively influenced the availability of nutrients (Mg and B) for the cocoa crop. The presence of Ca in the soil could be correlated with earthworm abundance. The low earthworm populations in monoculture highlight the need to evaluate more sustainable agricultural production approaches in the Ecuadorian Amazon in order to restore biodiversity and mitigate the loss of the soil's fauna.

Recent research highlights the potential of agroforestry systems to achieve sustainable and climate change-adapted agriculture. This study contributes to this research through evidence-based results. Overall, the effectiveness of agroforestry can be demonstrated by understanding the independent and combined effects involved in these agricultural production systems.

**Supplementary Materials:** The following supporting information can be downloaded at: https://www.mdpi.com/article/10.3390/f15010195/s1, Table S1: Mean values for fruit yield determined for the interaction Agroforestry systems × Production cycle and Table S2: Mean values for stored C determined for the interaction Agroforestry systems × Production cycle.

**Author Contributions:** Conceptualization: L.T.-J. and Y.V.-T.; methodology: L.T.-J., Y.V.-T. and A.C.; statistical analysis: N.H. and M.A.; writing—original draft preparation: L.T.-J., Y.V.-T., W.V., Y.V.-T., W.V.-C., C.C. and F.P.-A.; writing—review and editing: W.V., Y.V.-T., L.T.-J., N.H. and W.V.-C. All authors have read and agreed to the published version of the manuscript.

**Funding:** This study was funded by the National Institute of Agricultural Research (INIAP) and the Research Fund for Agrobiodiversity, Seeds, and Sustainable Agriculture (FIASA) in Ecuador. Project number 006. This project received funding from the European Union's Horizon 2020 MSCA-RISE 2019 programme under grant agreement 872384.

**Data Availability Statement:** Data is contained within the article.

**Acknowledgments:** We thank the agronomists of the EECA and Sustainable Agriculture (FIASA) in Ecuador.

**Conflicts of Interest:** The authors declare no conflict of interest.

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
