# Peer review of "Agroforestry Systems of Cocoa (Theobroma cacao L.) in the Ecuadorian Amazon"

_forests, doi:10.3390/f15010195_

Round 1

Reviewer 1 Report

Comments and Suggestions for Authors

Author Response

The response to the reviewer is in the PDF attached.

Reviewer 2 Report

Comments and Suggestions for Authors

Agroforestry systems of cocoa (Theobroma cacao) in the Ecuadorian Amazon

 The article highlights the importance of cocoa production in Ecuador and the negative environmental impacts of cocoa monocultures. It introduces agroforestry systems (AFSs) as a sustainable alternative to reduce greenhouse gas emissions and improve crop productivity. The results showed that agroforestry systems had better cocoa yield and total stored carbon compared to monocultures.

Despite the findings, here are a few potential limitations to note:

Major Issues

 The study is adequate for this publication; however, the language is inconsistent, using several words to explain the same idea at times, repeating phrases in paragraphs, and making certain grammatical errors. Writing should be checked by a native English speaker with agricultural knowledge as a good editorial practice.

 For a wider comprehension by English speakers, the majority of the paragraphs should be summarized.

A full An evaluation of the agroforestry system is required. This covers the economic and environmental features generated by the adjacent species in order to determine the true worth of the AFS and compare it to cocoa monoculture.

 Introduction

Lines: 52-71: Redo the paragraph, taking into account that most parentheses should be removed and a complete sentence written.

The introduction has many repetitive words and ideas; a thorough revision of the A change in wording is required.

At the end, of this session, author should write a paragraph that includes the objectives and hypotheses they developed.

 Materials and Methods

Line 119. Delete “However, only” start with... 36 central plants were…

Place all of the information on the species used, distance, use, and crown type, among other things, in a table.

Lines 143 – 144; 150 - 151. Where appropriate, move the results placed in this session.

Line 168: Review the word “in palm” for coherency.

Lines 168 – 176: Generally, a word should be used no more than once in a paragraph, as demonstrated by the three times the word "carried out" appears. Aim to keep this warning throughout the article.

 Results

 This session should be organized using the order of appearance of the variables in the methodology, it is advisable to use subtitles to separate the ideas and findings. In the same way, it is not a best editorial practice to start paragraphs with tables and figures; instead, the results should be presented so that the most significant findings are presented first, backed up by statistical evidence.

 Line 276. Move the sentence to methods.

 Lines 277 -278: The sentence seem to be wordy, summarize it. For example: “Cocoa production cycles had a highly significant effect for interaction.”

 Lines 292 – 293. What does the colors mean in Figure 2?

 Lines 310 – 312: Redo the paragraph.

 All the results must be supported by the statistical evidence, please put some letters above all numbers in the tables to the identify it.

 Line 370. Between correlation and regression, there is some misunderstanding. If that is the case, place the results appropriately.

 Discussion

 It should be noted in this section that the authors cited contribute to the discussion of the results. The authors should try to end each paragraph by incorporating all the ideas and contrasting them with the findings of this study. As a result, each paragraph should attempt to explain the implications of the obtained results.

 Conclusions

Conclusions should be focused on the hypotheses and objectives set out using the statistical results.

Comments on the Quality of English Language

The article should be proofread by a native English speaker and improve its presentation and coherence.

Author Response

(The authors gave the same response as above.)

Reviewer 3 Report

Comments and Suggestions for Authors

The current investigation entitled “Agroforestry systems of cocoa (Theobroma cacao) in the Ecuadorian Amazon” authored by Tinoco-Jaramillo et al., aimed to examine the influence of agroforestry systems on crop yield, carbon sequestration, the presence of earth- worms, and the nutritional contribution of companion species associated with cocoa (Theobroma cacao) cultivation under agroforestry systems. The finding of the current investigation indicated that the inclusion of companion species in agroforestry systems has a positive influence on cocoa productivity in the Ecuadorian Amazon

Comments/suggestions

The author should provide more detail about the results in the abstract especially in line 25-27 with some quantitative data. Otherwise, the abstract is well written.

In the introduction section, the background information is comprehensively provided. Previous investigations are also well discussed. However, the authors didn’t provide any information regarding the research gap of the current investigation. Moreover, the primary as well as the specific objectives or hypothesis of the current investigation are also missing from the manuscript.

In the material and method section, subsection 2.1. Also provide the information about the edaphic characteristics including soil properties and topographic details such as aspect, slope etc. The figure 1. is well presented.

In line 174. The authors have mentioned that weed control was carried out using herbicide; however, authors need to mention which herbicide are used along with quantity. Similarly, I did not find any detail about the irrigation scheduling for the crop.

The authors have indicated that “The determination of N, P, K, Ca, Mg, S, B, and Zn was carried out following the modified Olsen method,” but this is not possible since for the estimation of different soil nutrients, different procedures are used. For instance Olsen method is used for available P whole kjeldahl method for Available N. Wakley and Black method for organic carbon. So authors should check subsection 2.4.1. Section 2.4.2 the title of the subsection should be checked.

Figure 2. the author should also provide the standard deviation in the form of error bars for better clarity.

The soil properties data should be provided for each variation. Moreover in table 12, the author have indicated stepwise regression. However no regression equation provide. Moreover, it will be better, if author can provide correlation matrix as figure for better  clarity. Along with significance level. Simultaneously, the unnecessary tables such as Table 10, 12 and so one should be shifted to the supplementary file.

Discussion  section needs to be strengthen and particularly author need to focus on the cause and effect of the  finding of the current investigation. Moreover, the discussion should be under the same sub-section of the results for better understanding to readers. Moreover, more recent reference should be added. The  small paragraph should be avoided. In the conclusion section, the construckti9ve statements should be provided and I did not find any data related to the soil properties in the manuscript. Authors need to include it in the manuscript. At the end of conclusion section, the limitations of the current investigation and way forwards also need to be mentioned.

Comments on the Quality of English Language

 Moderate editing of English language required

Author Response

(The authors gave the same response as above.)

Round 2

Reviewer 1 Report

Comments and Suggestions for Authors

good job

Author Response

Comment: Good job.

Response: The authors thank the reviewer for her/his suggestions to improve the quality of this manuscript.

Reviewer 3 Report

Comments and Suggestions for Authors

In the first round of revision, the author have made considerable corrections in the manuscript. However, there are still minor suggestion which authors need to be taken care off such as 

The author have provided the primary objective of the investigation, however, authors also need to provide the specific objective /hypothesis of the current investigation. 

Moreover, i also suggest author to go for the economic analysis of the agroforestry for putting more strengthen on your suggestion. 

Regards

Comments on the Quality of English Language

 Minor editing of English language required

Author Response

Comment: The author have provided the primary objective of the investigation, however, authors also need to provide the specific objective /hypothesis of the current investigation. 

Response: This correction has been done in the last paragraph of the introduction section.

Comment:  Moreover, i also suggest author to go for the economic analysis of the agroforestry for putting more strengthen on your suggestion. 

Response: The benefit-cost ratio value for the agroforestry system which showed the highest yield and for the monoculture has been added as information in the discussion section. In addition, it has been written as a recommendation to carry out a deeper economic analysis over time.